# Enhancing Performance of Multilayer Perceptrons by Knot-Gathering Initialization

## Abstract

Multilayer perceptrons (MLPs) with `ReLU`-like activation functions form a high-dimensional, piecewise linear function space characterized by "knots"—points of non-differentiability. The density of such knots within a given input domain measures the MLP's capacity for function approximation. Despite the simplicity of this concept, knots remain underexploited in enhancing the practical performance of MLPs. This paper introduces Knot Gathering Initialization (KGI), a novel method that amplifies the local expressiveness of MLPs by increasing the knot density within the input domain prior to training. As an initialization technique, KGI is lightweight, data-independent, and hyperparameter-insensitive. The concept of knots, and hence KGI, can be directly generalized to smooth activation functions from different angles, including geometry, information transmission, and spectral analysis. We demonstrate the effectiveness of KGI across diverse tasks, including curve and surface fitting, image classification, time series regression, physics-informed operator learning, representation disentanglement, and large language model pretraining. These experiments unexceptionally show that KGI improves both the accuracy and convergence speed of MLPs, whether used standalone or as components of larger architectures. Promising future directions include: 1) the natural extension of KGI to convolutional and graph convolutional layers, as well as Low-Rank Adaptation for finetuning; and 2) applying knot gathering throughout training, rather than just at initialization.

## 1 Introduction

A neural network represents a high-dimensional, nonlinear function parameterized by its weights. Training involves optimizing these weights to approximate a target function or distribution in terms of a loss function. The ability of a neural network to approximate complex functions is referred to as its expressive power, or *expressiveness* (Raghu et al., 2017; Bengio & Delalleau, 2011). While expressiveness can be largely inferred from the number of weights, more advanced theories—such as the universal approximation theorems (Hornik, 1991), capacity measures (Shalev-Shwartz & Ben-David, 2014), and the concept of knots in piecewise linear networks (Montufar et al., 2014)—provide deeper insights, which will be discussed further in the Related Work section. The simplest

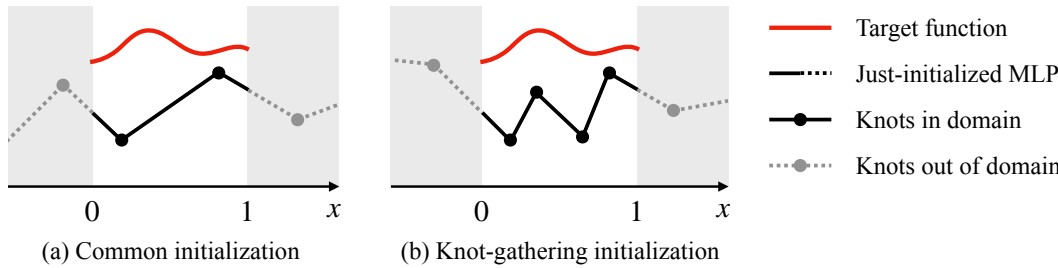

(a) Common initialization      (b) Knot-gathering initialization

Figure 1: Schematic of KGI. This method aims to initialize MLPs with a higher density of knots within the input domain (illustrated as $[0, 1]$). It enhances local expressiveness before training, leading to improved accuracy and faster convergence.

way to elevate expressiveness is to expand network size, but this approach requires more data and computational resources for training. Alternatives may include improving network architectures, activation functions, and loss functions. Regularization techniques, on the other hand, aim to control expressiveness to balance approximation and estimation errors (i.e., to mitigate overfitting), such as dropout (Srivastava et al., 2014) and weight decay (Loshchilov & Hutter, 2019).

A key consideration in expressiveness is the input domain. For instance, if the input $x \in [0, 1]^m$, the neural network is expected to be sufficiently expressive *only* within the hypercube $[0, 1]^m$. This notion of *local expressiveness* is critical for training dynamics and is closely tied to techniques that stabilize the data flow through the network, such as weight initialization, weight normalization, and hidden state normalization (Glorot & Bengio, 2010; Salimans & Kingma, 2016; Ioffe & Szegedy, 2015). Beyond model improvements, local expressiveness can also be enhanced by partitioning the input domain for parallel training (Meng et al., 2020). These approaches will be further discussed in the Related Work section. While they have demonstrated practical effectiveness, their underlying mechanisms often remain implicit. This paper aims to explicitly and interpretably harness local expressiveness to improve network performance.

We are inspired by two key ideas: the quantification of knots in multilayer perceptrons (MLPs) (Montufar et al., 2014), and the Kolmogorov-Arnold Networks (KANs) (Liu et al., 2023). Neural networks with piecewise linear activation functions, such as ReLU, inherently represent piecewise linear functions. Pascanu et al. (2013) and Montufar et al. (2014) were the first to quantify the total number of knots in MLP, conveying that knots can serve as a local measure of expressiveness, although they did not estimate such numbers within a compact set. KAN, an innovative architecture based on the Kolmogorov-Arnold representation theorem (Kolmogorov, 1957), features learnable activation functions on the edges between neurons. In KANs, each edge defines a univariate function anchored by a set of B-spline bases, whose arrangement (location and density) dynamically adjusts to the input range, thereby localizing expressive power. In contrast, knot distributions in MLPs remain unpredictable, which may explain why KANs achieve higher accuracy with fewer parameters in low-dimensional tasks.

Combining these two ideas, we aim to improve MLP performance by increasing the number of knots within the input domain, as illustrated in Figure 1. We are generalizing the concept of knots to smooth activation functions (such as GELU and Tanh) is straightforward but rigorous. In this paper, we focus on the weight initialization stage, introducing a method we call Knot-Gathering Initialization (KGI). As an initialization technique, KGI introduces no overhead to training and is compatible with all previously mentioned techniques. To ensure a thorough evaluation, we focus on MLPs in this study. In Discussion section, we will explore KGI's potential extension to the training stage, as well as its applicability to convolutional and graph convolutional neural networks (CNNs and GCNs; LeCun et al., 1998; Kipf & Welling, 2017), and parameter-efficient finetuning with Low-Rank Adaptation (LoRA; Hu et al., 2022). The main contributions of this paper include:

1. The concept of amplifying local expressiveness by gathering knots into the input domain.
2. A lightweight implementation of this concept through weight initialization, with significant and interpretable effectiveness.

The remainder of this paper details our method, starting with an in-depth literature review, followed by our methodology and experimental results, and concluding with discussions on broader implications and future directions.

## 2 RELATED WORK

The central idea of this paper is to enhance the local expressiveness of neural networks through weight initialization. In this section, we review fundamental theories on expressiveness and methodologies for its enhancement, focusing on the role of knots and initialization techniques.

### 2.1 EXPRESSIVENESS OF NEURAL NETWORKS

**Universal approximation** First established by Cybenko (1989) and expanded by Hornik (1991), the Universal Approximation Theorem (UAT) underpins the theoretical capability of neural networks. Given sufficient neurons, it asserts that an MLP with a single hidden layer can approximate

any continuous function on a compact set. Further studies have extended UAT to deep networks, showing the influence of network depth and activation functions (Lu et al., 2017; Hanin & Sellke, 2019). UATs have also been proven for specialized architectures such as ResNet (Lin & Jegelka, 2018) and DeepONet (Lu et al., 2021). Nevertheless, UATs do not quantify expressiveness due to unbounded approximation error and are not concerned with estimation error or generalization.

**Capacity measures**    Various capacity measures have been proposed for a more quantified understanding of expressiveness. The earliest and most fundamental measure could be the VC dimension (Vapnik & Chervonenkis, 2015), which counts the maximum number of data points a model can shatter or perfectly classify. Based on the VC dimension, the PAC (probably approximately correct) learning theorems provide a framework to constrain the generalizability of learning algorithms (Shalev-Shwartz & Ben-David, 2014). Extensions such as the fat-shattering and shattering coefficients generalize the VC dimension to real-valued functions and distinct labelings (Kearns & Schapire, 1994; Anthony et al., 1999). Beyond these combinatorial measures, Rademacher complexity offers a data-dependent perspective, assessing how well a model can fit random noise, with lower values indicating better generalizability (Bartlett & Mendelson, 2002). Last, the tensor rank of layers, particularly in networks with multiplicative interactions, reflects its capacity to model complex, multilinear interactions (Cohen et al., 2016).

**Linear regions or knots**    In piecewise linear networks, such as those using ReLU activation functions, the input space is partitioned into regions where the network behaves as a linear function. The number of these linear regions, or knots between them, correlates with the network's capacity to represent complex functions. This thought may date back to Pascanu et al. (2013) and Montufar et al. (2014) who, respectively, found the upper and lower bounds of this number for MLPs, considering impacts of width, depth, and activation functions. Further studies have explored the combinatorial structure of knots in different architectures, such as CNNs and networks with varying types of activation functions (Serra et al., 2018; Arora et al., 2018). More recent work has provided dynamic analyses of how these knots evolve during training and how architectural choices influence their number and distribution (He et al., 2018; Hanin & Rolnick, 2019). These insights are crucial to understanding the network design trade-offs and optimizing the expressiveness of deep learning models (Raghu et al., 2017; Bengio & Delalleau, 2011).

Knots offer distinct advantages over other capacity measures due to their locality and simplicity. First, they are a local measure that enables the assessment of a network's capacity within specific input domains. Second, the position of knots can be directly linked to the network's weights and biases without relying on more abstract theoretical concepts (e.g. shattering). These properties make knots a pragmatic tool for quantifying and enhancing model performance, which inspires our work.

## 2.2 ENHANCING LOCAL EXPRESSIVENESS

**Weight initialization**    Weight initialization is critical for training neural networks, significantly affecting both convergence speed and final model performance (Narkhede et al., 2022). Early approaches, such as random initialization, were prone to vanishing or exploding gradients, especially in deep networks (Bengio et al., 1994). To mitigate these issues, methods such as Xavier (Glorot & Bengio, 2010) and He initialization (He et al., 2015) were introduced, scaling weights based on input and output sizes to maintain stable activation throughout the network, effectively preventing diminished expressiveness in the input domain. These techniques have since become standard practice and will serve as the starting point of our knot-gathering process. Recent advances have continued to refine these methods, particularly in the context of deeper and more complex architectures. For instance, Mishkin & Matas (2016) introduced the initialization of the Sequential Unit Variance (LSUV), which processes the pre-initialized weights layer by layer to further normalize output variance. Hanin & Rolnick (2018) highlighted the critical interplay between weight initialization and modern normalization techniques. Focusing on transformers, Zhu et al. (2021) presented a dynamic approach that learns optimal initial weights during training to enhance stability and improve convergence. Our method, KGI, shares similar characteristics: layer-wise processing, an emphasis on the interplay between layers, and compatibility with training. However, driven by knot positioning, we deem KGI to have the most interpretable mechanism.

**Normalization**    Normalization can also be understood as localizing the expressive power by constraining data flow activation (Huang et al., 2023). It can be applied to hidden states across batch (Ioffe & Szegedy, 2015), feature (Ba et al., 2016), and group (Wu & He, 2018) dimensions, or directly to weights (Salimans & Kingma, 2016; Qiao et al., 2019). Although not the focus of this work (which concerns initialization), the real-time tracking of post-activation distributions through normalization can facilitate the extension of knot gathering to the training process, as explained in the Discussion section.

**Dynamic activation**    Just as data can be adapted to activation through normalization, activation functions can also be adapted to data, giving rise to adaptive or trainable activation functions (Dubey et al., 2022). Some adaptive functions, such as Swish and cubic splines, learn their shapes based on heuristic insights (Ramachandran et al., 2017; Scardapane et al., 2019). However, many others have formally incorporated the positioning of critical points, such as APL (Adaptive Piecewise Linear), BDAA (Bi-modal Derivative Activation), linear regression-based activation training, Mexican `ReLU`, and Gaussian radial basis functions (RBFs) (Agostinelli, 2014; Mishra et al., 2017; Ertuğrul, 2018; Maguolo et al., 2021; Jiang et al., 2021). The concept of critical point positioning is made more explicit in KAN (Liu et al., 2023; Yu et al., 2024), where the support of activation functions dynamically adjusts to that of data—another key inspiration for this work, as discussed in the Introduction section. Our method stands out by achieving critical point positioning through weight adjustments, rather than altering activation functions, making it naturally compatible with commonly-used neural network layers and introducing no overhead to training.

**Domain decomposition**    The techniques discussed above focus on enhancing the local expressiveness of a single neural network. Domain decomposition, by contrast, adopts a different perspective: training multiple neural networks in parallel, each responsible for a sub-region of the input domain. This method has proven effective for training physics-informed neural networks (PINNs) to emulate or invert partial differential equations (PDEs; Meng et al., 2020; Shukla et al., 2021; Moseley et al., 2023), where the underlying functions are too complex for a single global model to capture. The concept of domain decomposition reinforces our insights into local expressiveness and can directly benefit from our approach, which explicitly accounts for sub-domain supports.

## 3    KNOT GATHERING INITIALIZATION

We detail our methodology in this section, beginning with a single layer using `ReLU` activation and progressively extending it to MLPs and smooth activation functions.

### 3.1    SINGLE LAYER

Consider a linear layer with weight $\boldsymbol{W} \in \mathbb{R}^{n \times m}$ and bias $\boldsymbol{b} \in \mathbb{R}^n$, mapping input $\boldsymbol{x} \in [a, b]^m$ to output $\boldsymbol{y} \in \mathbb{R}^n$ by

$$\boldsymbol{y} = \texttt{ReLU}(\boldsymbol{z}), \quad \text{where} \quad \boldsymbol{z} = \boldsymbol{W}\boldsymbol{x} + \boldsymbol{b}. \tag{1}$$

Assume that the weight $\boldsymbol{W}$ and bias $\boldsymbol{b}$ are properly pre-initialized to $\boldsymbol{W}_0$ and $\boldsymbol{b}_0$, for instance, using He uniform or normal initialization (He et al., 2015). *The goal of KGI is to modify $\boldsymbol{W}_0$ and $\boldsymbol{b}_0$ such that the knot of `ReLU`, or the root of $\boldsymbol{z} = \boldsymbol{0}$, lies within the input domain $[a, b]^m$.*

Let $\hat{\boldsymbol{x}}$ represent the root of $\boldsymbol{z} = 0$ (e.g., $\hat{\boldsymbol{x}} = -\boldsymbol{W}^{-1}\boldsymbol{b}$ when $m = n$). Directly adjusting $\boldsymbol{W}_0$ and $\boldsymbol{b}_0$ to ensure that $\hat{\boldsymbol{x}}$ lies within $[a, b]^m$ is challenging, particularly in high dimensions. However, if we know the root $\hat{\boldsymbol{x}}$, it is straightforward to adjust either the weight $\boldsymbol{W}_0$ or the bias $\boldsymbol{b}_0$ while keeping the other fixed. Thus, we adopt a "knot-sampling and weight-perturbation" strategy.

We first sample an $\hat{\boldsymbol{x}}$ from $[a, b]^m$:

$$\hat{\boldsymbol{x}} \sim \mathcal{U}(a, b)^m. \tag{2}$$

Given this $\hat{\boldsymbol{x}}$, two approaches are evident:

1. **Bias-Modifying Approach:**    Fixing the weight at $\boldsymbol{W} = \boldsymbol{W}_0$, the bias is modified to

$$\boldsymbol{b}_{\text{KGI}} = -\boldsymbol{W}_0\hat{\boldsymbol{x}}. \tag{3}$$

2. **Weight-Modifying Approach:** Fixing the bias at $\boldsymbol{b} = \boldsymbol{b}_0$, the weight is modified to

$$\boldsymbol{W}_{\text{KGI}} = \lambda \boldsymbol{W}_{\text{h}} + \boldsymbol{W}_{\text{p}}, \tag{4}$$

where $\boldsymbol{W}_{\text{h}}$ is the homogeneous part satisfying $\boldsymbol{W}_{\text{h}}\hat{\boldsymbol{x}} = 0$, and $\boldsymbol{W}_{\text{p}}$ is the particular part satisfying $\boldsymbol{W}_{\text{p}}\hat{\boldsymbol{x}} + \boldsymbol{b}_0 = 0$:

$$\boldsymbol{W}_{\text{h}} = \boldsymbol{W}_0 - \frac{\boldsymbol{W}_0 \hat{\boldsymbol{x}} \hat{\boldsymbol{x}}^{\text{T}}}{\hat{\boldsymbol{x}}^{\text{T}}\hat{\boldsymbol{x}}}, \quad \boldsymbol{W}_{\text{p}} = -\frac{\boldsymbol{b}_0 \hat{\boldsymbol{x}}^{\text{T}}}{\hat{\boldsymbol{x}}^{\text{T}}\hat{\boldsymbol{x}}}, \tag{5}$$

and $\lambda$ represents any real number and is treated as a hyperparameter. In most cases, setting $\lambda = 1$ is sufficient.

A caveat of using Eqs. (3) or (4) is that all pre-activation outputs, $z_1(\boldsymbol{x})$, $z_2(\boldsymbol{x})$, $\cdots$, $z_n(\boldsymbol{x})$, will share the same zero, i.e., the sampled $\hat{\boldsymbol{x}}$. To remove this algorithm-imposed restriction, distinct $\hat{\boldsymbol{x}}$ values can be sampled to compute each row of $\boldsymbol{b}_{\text{KGI}}$ or $\boldsymbol{W}_{\text{KGI}}$. However, we opt for a simpler approach: perturbing $\boldsymbol{W}_{\text{KGI}}$ and $\boldsymbol{b}_{\text{KGI}}$ by a fraction of their pre-initialized counterparts, yielding the final initialization as

$$\boldsymbol{W}_0^* = \alpha \boldsymbol{W}_0 + (1 - \alpha)\boldsymbol{W}_{\text{KGI}} \quad \text{or} \quad \boldsymbol{b}_0^* = \alpha \boldsymbol{b}_0 + (1 - \alpha)\boldsymbol{b}_{\text{KGI}}, \tag{6}$$

where $\alpha \in [0, 1]$ is a hyperparameter, referred to as the *perturbation factor*. When $\alpha > 0$, the knots may no longer be confined to $[a, b]^m$, introducing a trade-off between gathering knots and preserving randomness. Our experiments suggest that selecting $\alpha$ in the range $[0.1, 0.5]$ provides a good balance, with performance showing minimal sensitivity to the exact value within this interval. This leaves the domain bounds, $a$ and $b$, as the only influential hyperparameters for each layer.

In our experiments, we observed no significant difference in performance between the two approaches. When a bias-free layer is required, the weight-modifying approach can be employed. However, as we explain in the Discussion section, the bias-modifying approach is innately compatible with non-affine layers, such as those in CNNs and GCNs.

## 3.2 MULTIPLE LAYERS

For MLPs, we need to estimate the bounds $a$ and $b$ for each layer based on the activation function of its preceding layer. Heuristically, we recommend using $[0, 1]$ for `ReLU` and similar activations, $[-0.8, 0.8]$ for `Tanh`, and $[0.1, 0.9]$ for `Sigmoid`, as these ranges align with their corresponding function ranges. We avoid the full range of `Tanh` and `Sigmoid` because the extreme values are rarely reached in pre-initialized models. While these suggested ranges can be finetuned for better performance, they are generally sufficient to make KGI effective. It must be emphasized that our objective is to *increase*, rather than to maximize, knot density within the input domain: the perturbation factor $\alpha$ is intended to remove some of the knots from the input domain.

Alternatively, we provide an automated approach if a representative batch of data is available. For each layer, we calculate the minimum and maximum of its input, denoted as $d_{\text{min}}$ and $d_{\text{max}}$, and determine $[a, b]$ by

$$[a, b] = [d_{\text{min}} + \beta(d_{\text{max}} - d_{\text{min}}), d_{\text{max}} - \beta(d_{\text{max}} - d_{\text{min}})], \tag{7}$$

where $\beta \in [0, 0.5)$ creates a margin at both ends. Note that using the mean and variance of the input is inappropriate when the preceding activation is `ReLU` (since half of the inputs vanish). With $a$ and $b$ determined, KGI can be performed, followed by a forward pass to propagate the activated output to the next layer. This layer-by-layer approach is akin to LSUV (Mishkin & Matas, 2016). In practice, finding a truly representative batch can be difficult in a high-dimensional space, so we recommend using the heuristic estimates to ensure stable training dynamics.

## 3.3 SMOOTH ACTIVATION FUNCTIONS

The concept of a knot can be generalized to smooth activation functions as where they are most expressive. For example, in smooth variants of linear units, such as `ELU`, `SELU`, and `GELU` (Clevert et al., 2016; Klambauer et al., 2017; Hendrycks & Gimpel, 2017), as well as functions like `Swish` and `Mish` (Ramachandran et al., 2017; Misra, 2020), $z = 0$ can naturally serve as the knot. For more generic activation functions, we provide three consistent perspectives: geometry, information transmission, and spectral analysis.

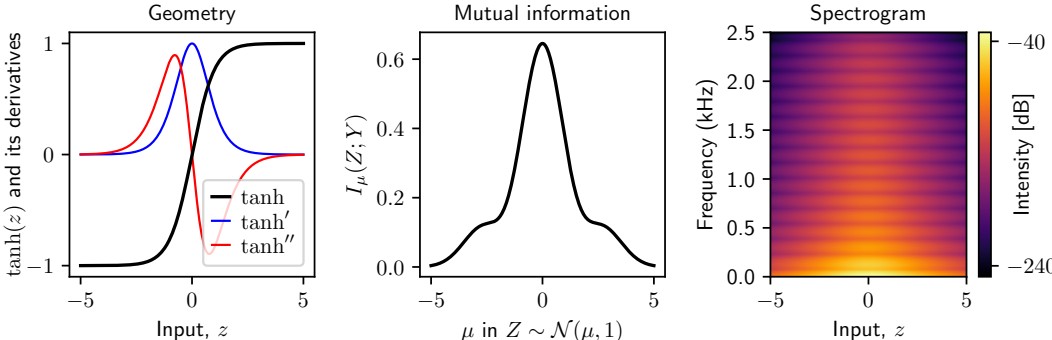

Figure 2: Evidence for $z = 0$ as a "knot" in `Tanh`. The mutual information $I_\mu(Z;Y)$ is determined by Eq. (13), assuming $Z \sim \mathcal{N}(\mu, 1)$ and $Y = \tanh(Z)$. The spectrogram is computed with the spatial and frequency domains both discretized into 5000 points.

Consider `Tanh` as an example. We state that $z = 0$ is its knot based on the following evidence (illustrated in Figure 2):

1. Geometrically, $z = 0$ is an inflection point maximizing the slope, indicating that the function is most sensitive to changes in input around $z = 0$.

2. When random input $Z \sim \mathcal{N}(\mu, 1)$ is passed through $Y = \tanh(Z)$, the mutual information (MI) (MacKay, 2003) between $Z$ and $Y$ is maximized at $\mu = 0$, as detailed in Appendix A. This suggests that $z = 0$ is an optimal point for information transmission.

3. The spectrogram of `Tanh` (from Short-time Fourier transform) shows that high-frequency components are concentrated near $z = 0$.

Such analyses can be applied to commonly used activation functions, suggesting that $x = 0$ often serves as a natural knot. In cases where the knot is non-zero in a specific activation function, we can adjust the pre-initialized bias $\boldsymbol{b}_0$ by subtracting the knot, ensuring that Eqs. (3) and (4) still hold.

## 4    EXPERIMENTS

We conducted seven experiments across different domains and model scales. In this section, we will first present results from two low-dimensional problems, highlighting the mechanism of KGI via knot visualization. Following this, we will report on the remaining high-dimensional problems.

Since KGI introduces a mechanism distinct from previous studies and is compatible with most existing techniques (as reviewed in Section 2.2), we do not use a static baseline in our experiments. Instead, the baseline, as labeled "No KGI" in the subsequent sections, is tailored to each problem, incorporating standard practices (such as pre-initialization, normalization, and regularization) to achieve reasonable performance without KGI. To demonstrate KGI's robustness and usability, we do not tune its hyperparameters; we simply estimate the bounds $[a, b]$ from the preceding activation functions, fixing the homogeneous factor $\lambda = 1$ and perturbation factor $\alpha = 0.2$.

### 4.1    CURVE AND SURFACE FITTING

We train MLPs to fit a curve and a surface, as shown in Figures 3a and 3c, both characterized by spiky shapes that require localized expressive power. We explore ten architecture scales, varying in width $W$ (number of units per layer) and depth $D$ (number of hidden layers, excluding the input and output layers), and four activation functions: `ReLU`, `LeakyReLU`, `GELU`, and `Tanh`. The models are trained using mean squared error (MSE) as the loss function and Adam as the optimizer, with a learning rate of $10^{-3}$. Training is performed for 30,000 epochs on unbatched data. For KGI, we use the weight-modifying approach, setting the input domain ($[a, b]$) to $[-0.8, 0.8]$ for `Tanh`, and $[0.2, 0.8]$ for the other activation functions.

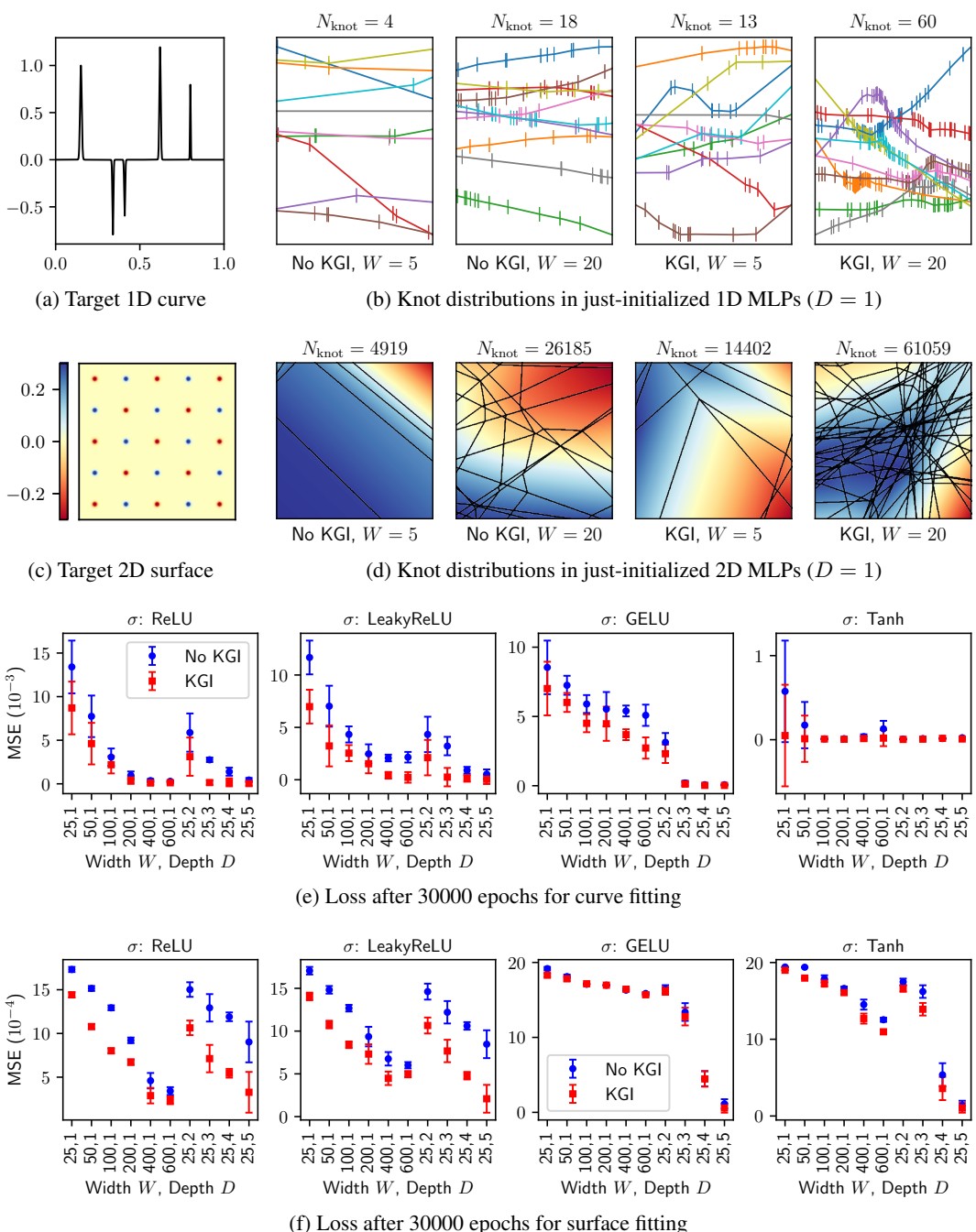

Figure 3: Fitting a curve and a surface with MLPs. **(a) and (c)** show the targets with Gaussian bell-shaped spikes, discretized into 1,000 and 200×200 data points, respectively. **(b) and (d)** display in-domain knot distributions in the just-initialized (untrained) MLPs using `ReLU` activation, with $W$ and $D$ representing network width and depth. The "No KGI" panels show models pre-initialized with He uniform (He et al., 2015), which are further processed by KGI and shown in the "KGI" panels. Each scenario includes ten model samples, with $N_{\mathrm{knot}}$ denoting the average number of knots within the input domains. In (b), all ten model samples are shown, with knots marked by vertical bars. In (d), a single model sample is visualized, with knots identified at pixels where significant horizontal or vertical slope changes occur, marked by small dots forming line segment patterns. **(e) and (f)** compare final losses for models initialized with and without KGI, accompanied by the loss histories shown in Figures 5 and 6.

Before presenting the training results, we visualize the knot distributions in untrained `ReLU`-based models, as shown in Figures 3b and 3d. It is shown that, for the same architecture, KGI increases the knot density within the input domain approximately by three times. A higher density of knots tends to accelerate convergence and improve the attained local minima, rendering KGI's mechanism more interpretable compared to many other methods in enhancing MLP performance.

The training results are presented in Figures 3e and 3f, reporting the final MSE for the different network sizes and activation functions, accompanied by the training histories visualized in Appendix B. Based on these results:

1. For smaller models, KGI consistently improves both convergence speed and final loss.

2. For larger models that are sufficiently sized for the tasks, the improvement in final loss may become negligible, but the improvement in convergence speed remains significant.

In addition, animations of knot distributions during training are provided in our code repository, visually illustrating how knot gathering improves training dynamics from start to finish. Refer to our Reproducibility Statement.

## 4.2 HIGH-DIMENSIONAL PROBLEMS

This section presents five high-dimensional experiments across various domains, tasks, data types, and network scales. We briefly describe each problem below, with further details on datasets, architectures, hyperparameters, and metrics provided in Appendix C.

1. **CIFAR-10**  A benchmark dataset for image classification (Krizhevsky, 2009). The metric is test accuracy. This experiment evaluates KGI's performance on a high-dimensional input domain ($[0,1]^{32 \times 32 \times 3}$).

2. **Element analysis**  A probability regression problem on spectral data, where the goal is to predict the percentage composition of elements in a sample based on its emitted muonic X-rays (Hillier et al., 2022; Cataldo et al., 2022). The primary metrics include KL-divergence (as the loss function) and threshold accuracy. This experiment highlights the challenges posed by real-world scientific data.

3. **Stokes Flow**  Solving the 2D Stokes flow in a square box with a moving lid using a Physics-Informed Deep Operator Network (PI-DeepONet; Lu et al., 2021). The metric is the relative L2 error compared to the ground truth. This problem introduces two challenges: the architecture consists of two MLPs—a branch net processing physical parameters and a trunk net handling spatial coordinates; besides, the loss function involves higher-order derivatives with respect to the input coordinates.

4. **Disentanglement**  Disentangling features with autoencoder-based models is highly sensitive to model initialization due to the non-convexity of disentangling losses (Locatello et al., 2019). We use variational autoencoders (VAEs) to learn disentangled features on the XY dataset (Cha & Thiyagalingam, 2023), with Joint Entropy Minus Mutual Information Gap (JEMMIG; Do & Tran, 2020) to measure the alignment between latent variables and ground-truth factors. We aim to evaluate whether KGI can stabilize training and improve disentanglement quality.

5. **GPT-2**  Pretraining GPT-2 (Radford et al., 2019) for causal language modeling. While this task typically requires a large dataset, such as OpenWebText with eight million documents (Gokaslan & Gao, 2019), we use WikiText-2 with ~40,000 documents (Merity et al., 2017) due to resource limitation. The metric is test accuracy. This experiment tests KGI's effectiveness in large-scale models, focusing on the early stages of training.

Alongside the problem-specific metrics mentioned above, we also report two common metrics across all experiments: the final loss after a sufficiently large number of epochs, and *convergence slowness*, which we define as the mean value of the loss curve divided by the initial loss. Geometrically, convergence slowness represents the relative area under the loss curve.

The metrics are reported in Table 1, with typical training histories illustrated in Figure 4. The results indicate that KGI consistently enhances model performance to varying extents, with the

Table 1: Losses and metrics for high-dimensional experiments.

| Problem | LOSS[1] | | SLOWNESS[2] | | METRIC | | |
| --- | --- | --- | --- | --- | --- | --- | --- |
| | No KGI | KGI | No KGI | KGI | Name | No KGI | KGI |
| CIFAR | 0.26±0.03 | **0.18±0.02** | 0.35±0.03 | **0.28±0.02** | Acc. | 46.65±0.31% | **47.53±0.27**% |
| Muon | 1.26±0.33 | **0.88±0.30** | 0.63±0.05 | **0.59±0.04** | Acc. | 88.51±3.66% | 90.06±4.16% |
| Stokes | 3.33±0.35 | **1.19±0.09** | 0.15±0.01 | **0.09±0.01** | Rel. L2 | 17.88±1.69% | **10.45±0.87**% |
| DisEnt[3] | **4.40±0.02** | 4.41±0.02 | 0.73±0.16 | **0.56±0.18** | JEMMIG | 0.62±0.14 | **0.71±0.10** |
| GPT-2 | 1.75±0.13 | **1.38±0.19** | 0.15±0.00 | **0.14±0.01** | Acc. | 20.56±0.18% | 20.89±0.21% |

[1] Unit of loss: $10^{-4}$ in Muon, $10^{-4}$ in Stokes, $10^2$ in DisEnt, and $10^{-2}$ in GPT-2.
[2] Unit of slowness: $10^{-2}$ in Muon.
[3] The loss for DisEnt does not capture the quality of feature disentanglement, as the reconstruction loss exerts an adversarial influence, often worsening disentanglement.

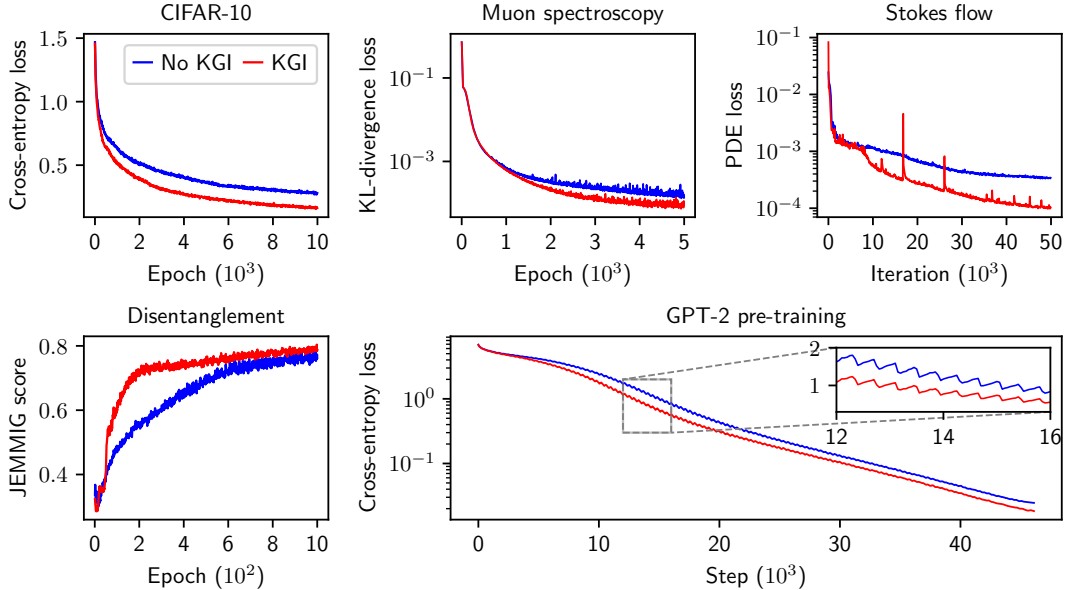

Figure 4: Training histories for high-dimensional experiments, with models initialized using the same random seed (i.e., not selectively chosen to favor KGI). The only exception is the Disentanglement case, where the best models with and without KGI are shown, as disentanglement may fail depending on initialization, often requiring multiple seeds (Do & Tran, 2020).

improvements on Stokes flow and disentanglement being particularly notable. Additional results and detailed analyses for each problem are provided in Appendix C.

## 5 DISCUSSION

The number of knots in piecewise linear neural networks has been explored as an intuitive and interpretable measure of their expressive power. We directly leverage this concept to enhance the practical performance of MLPs by increasing knot density within the input domain during weight initialization, a technique we refer to as Knot Gathering Initialization (KGI). KGI consists of three steps: sample a knot from the input domain, adjust the weight or bias to accommodate that knot, and perturb the weight or bias to diversify the knots across different output dimensions. This approach can be naturally extended to smooth activation functions, broadening the applicability of the knot concept. KGI consistently improves accuracy and convergence speed across various tested

real-world tasks, including image classification, time-series regression, physics-informed operator learning, representation disentanglement, and large language model pretraining.

## 5.1 ARCHITECTURAL AND TRAINING COMPATIBILITY

While this paper focuses on MLPs, KGI is formally compatible with non-affine architectures such as CNNs, GCNs, and LoRA (LeCun et al., 1998; Kipf & Welling, 2017; Hu et al., 2022). These architectures comprise layers of the form $z = Wx + b$ prior to activation, but with $W$ subject to specific structural constraints. For instance, in convolutional and graph convolutional layers, $W$ follows a fixed sparsity pattern with tied elements. In LoRA, $W$ takes the form $W = W_\mathrm{p} + AB$, where $W_\mathrm{p}$ is the pretrained weight, and $A$ and $B$ are unconstrained low-rank matrices for finetuning. Such prescribed weight structures preclude our weight-modifying approach (Eq. (4)) but still support our bias-modifying approach (Eq. (3)). Nevertheless, such formal compatibility does not guarantee KGI's effectiveness for these layers, which requires future investigation.

In addition to increasing knot density prior to training, we may also explore maintaining it throughout training. Knot gathering is also formally compatible with training if we treat $W_0$ and $b_0$ in Section 3.1 as the current weight and bias and apply knot gathering through Eq. (3) or (4) to regulate them. The mean and variance of the hidden state can be tracked during training, similar to normalization techniques (Ioffe & Szegedy, 2015; Ba et al., 2016; Wu & He, 2018), and used to dynamically sample the knot $\hat{x}$; as a particular case, if $\hat{x}$ is anchored to the tracked mean, knot gathering becomes deterministic, introducing no stochasticity. Potential side effects may include overfitting and unstable gradients. More experiments are needed to assess these issues.

## REPRODUCIBILITY STATEMENT

The code for this paper, including the implementation of KGI, Jupyter notebooks and datasets for running experiments, and training logs, will be made publicly available after the review process. For the review phase, the code (with git history removed) is included in the Supplementary Material. The Supplementary Material also contains animations of knot distributions during training, illustrating how knot gathering improves training dynamics.

Experiments involving stochastic processes were conducted with fixed pseudo-random seeds to ensure reproducibility. All figures and tables in this paper, except for Figure 1, can be reproduced with a single click with the provided training logs.

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

## A  INFORMATION TRANSMISSION THROUGH TANH

Sample $Z \sim \mathcal{N}(\mu, 1)$, i.e., a Gaussian random variable with mean $\mu$ and variance 1. With $Y = \tanh(Z)$, we aim to calculate the mutual information (MI) between $Z$ and $Y$.

The MI, as denoted by $I_\mu(Z; Y)$, is defined by (MacKay, 2003)

$$I_\mu(Z; Y) = H_\mu(Y) - H_\mu(Y|Z), \tag{8}$$

where $H_\mu(Y)$ is the entropy of $Y$, and $H_\mu(Y|Z)$ the conditional entropy of $Y$ given $Z$. Since $Y = \tanh(Z)$ is a deterministic transformation of $Z$, the conditional entropy $H_\mu(Y|Z)$ vanishes, simplifying $I_\mu(Z; Y)$ to $H_\mu(Y)$.

To determine $H_\mu(Y)$, we need the probability density function (PDF) of $Y$. The PDF of $Z$ is

$$p_Z(z) = \frac{1}{\sqrt{2\pi}} \exp\left(-\frac{(z-\mu)^2}{2}\right). \tag{9}$$

The derivative of $y = \tanh(z)$ is

$$\frac{dy}{dz} = 1 - \tanh^2(z) = 1 - y^2. \tag{10}$$

The PDF of $Y$ is given by the change of variables formula:

$$p_Y(y) = p_Z(z) \left|\frac{dz}{dy}\right|, \tag{11}$$

where $z = \tanh^{-1}(y)$. Substituting Eqs. (9) and (10) into (11), we obtain the explicit form of $p_Y(y)$:

$$p_Y(y) = \frac{1}{\sqrt{2\pi}} \exp\left(-\frac{(\tanh^{-1}(y) - \mu)^2}{2}\right) \frac{1}{1 - y^2}. \tag{12}$$

Note that the transformation $Y = \tanh(Z)$ compresses the real line into the interval $(-1, 1)$. Based on the definition of entropy, we eventually obtain

$$I_\mu(Z; Y) = H_\mu(Y) = -\int_{-1}^{1} p_Y(y) \log p_Y(y) \, dy. \tag{13}$$

Due to the complex form of $p_Y(y)$, analytical evaluation of the above integral is difficult. We compute it numerically, varying $\mu$ in interval $[-5, 5]$, and the results are displayed in the middle panel of Figure 2.

## B  TRAINING HISTORY FOR CURVE AND SURFACE FITTING

Complementing Figures 3e and 3f, Figures 5 and 6 present the training history for curve and surface fitting, respectively, across the considered network sizes and activation functions.

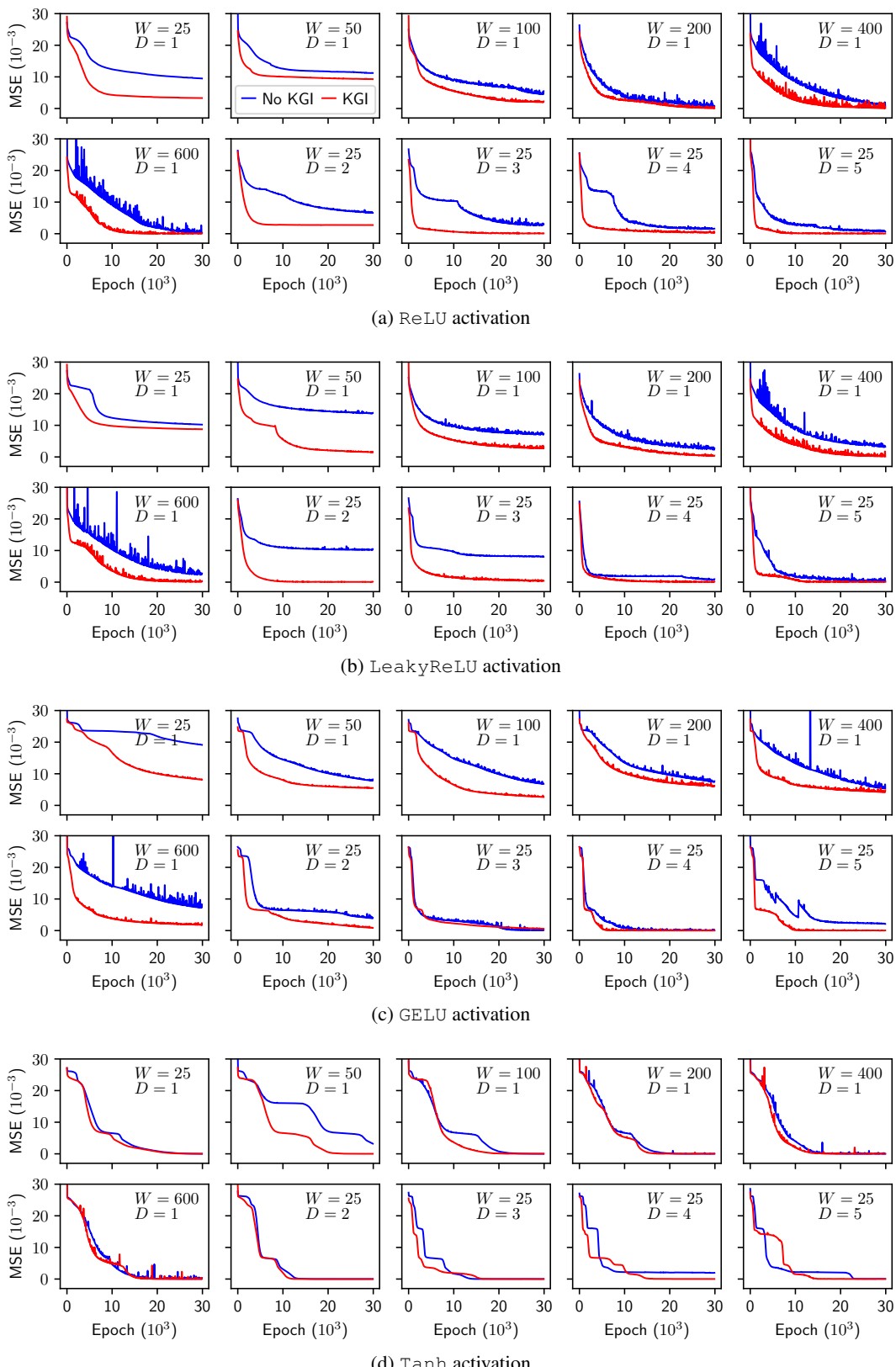

Figure 5: Loss histories in curve fitting. These models are initialized with the same random seed (i.e., not selected individually to favor KGI).

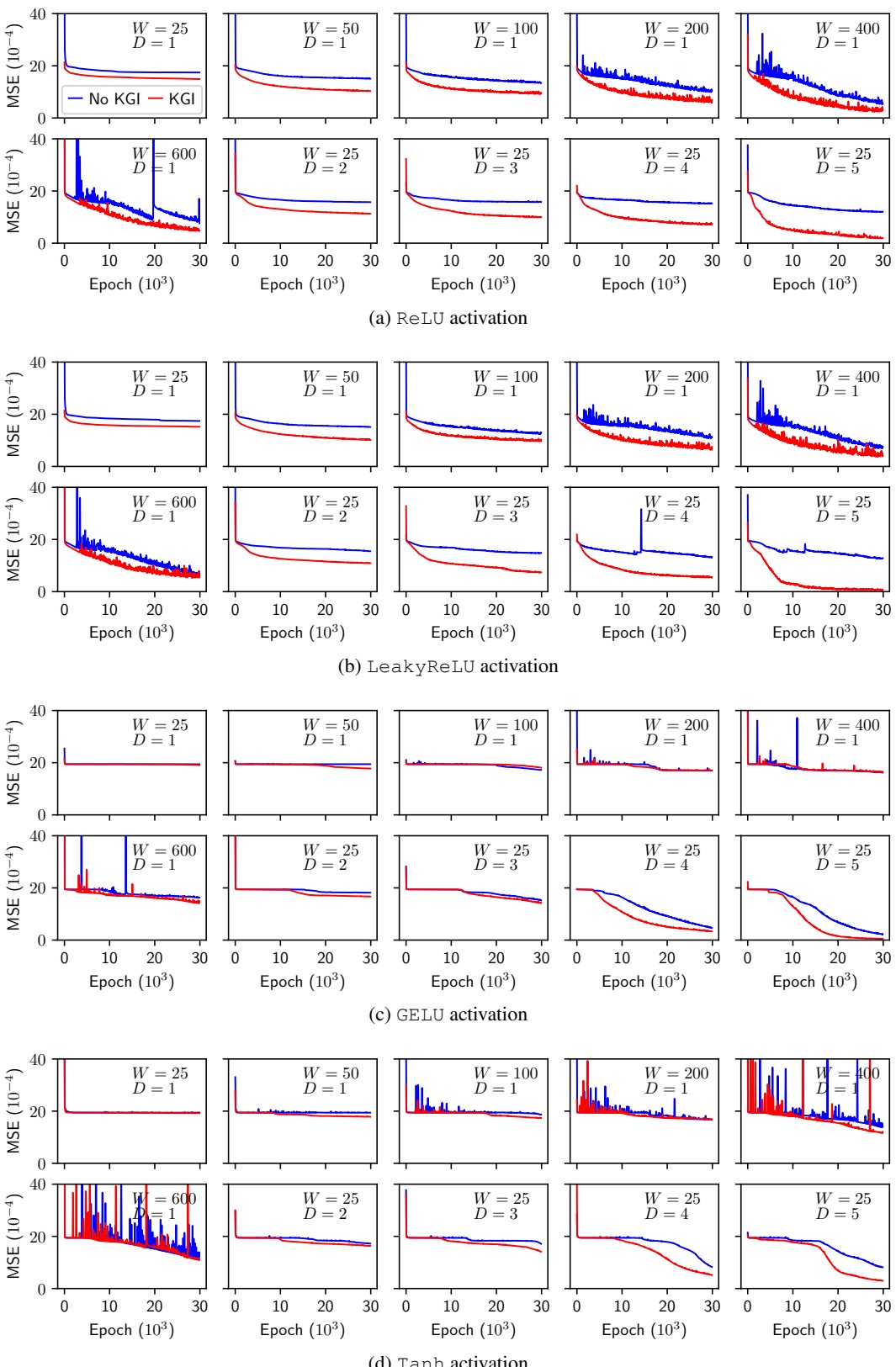

Figure 6: Loss histories in surface fitting. These models are initialized with the same random seed (i.e., not selected individually to favor KGI).

## C  DETAILS ON HIGH-DIMENSIONAL EXPERIMENTS

### C.1  CIFAR-10

CIFAR-10 (Krizhevsky, 2009) serves as a standard benchmark dataset for image classification, including 60,000 $32 \times 32$ color images, evenly distributed across 10 distinct classes. Our MLP comprises three linear layers with output sizes 256, 128, and 10, using `ReLU` activation. We employ cross entropy as the loss function and Adam as the optimizer, with a learning rate of $10^{-3}$. Training is conducted over 10,000 epochs with a batch size of 512. For KGI, we apply the weight-modifying approach over the interval $[a, b] = [0, 1]$, with the weights pre-initialized by He uniform. The results, presented in Table 1 and Figure 4, indicate that the MLP struggled with this task, achieving a test accuracy slightly below 50%. However, the usage of KGI led to accelerated convergence and improved accuracy.

### C.2  MUON SPECTROSCOPY

Muon spectroscopy is a powerful technique for elemental analysis (Hillier et al., 2022; Cataldo et al., 2022). Negative muons are implanted into a material and captured by its atoms, which then emit muonic X-rays as they relax in energy. These X-rays provide a unique fingerprint of the material's chemical composition, with their energy characteristic of the capturing atoms. The technique applies to a wide range of materials, including those in cultural heritage, energy technologies, advanced engineering, biomaterials, and green technologies. Quantifying elemental composition can be challenging due to spectral complexity.

In this experiment, we predict the elemental composition of samples from their muon spectra. The original dataset (Butler & Hillier, 2024) provides spectra with 4,000 energy points, spanning from 0 to 8,000 keV. To make the problem more challenging, we process the data by truncating the second half of the spectra and decimating the first half by a factor of two, resulting in 1,000 energy points per spectrum up to 4,000 keV. The output corresponds to the percentage composition of three elements: silver, gold, and aluminum. A representative spectrum is illustrated in Figure 7.

The MLP used in this experiment consists of three hidden layers, each with 128 units and `ReLU` activations. At the output layer, we apply `softmax` to normalize the predictions into probability distributions. In the KGI approach, weight modification is applied within the range $[a, b] = [0, 1]$, with the weights pre-initialized using He uniform. The model is optimized using Adam with a learning rate of $10^{-3}$ and trained for 5,000 epochs with a batch size of 512. KL-divergence serves as the loss function, while performance is evaluated using a 1% threshold accuracy metric—predictions are considered accurate if they fall within 1% of the ground truth. As shown in Table 1 and Figure 4, KGI noticeably accelerates convergence and improves accuracy.

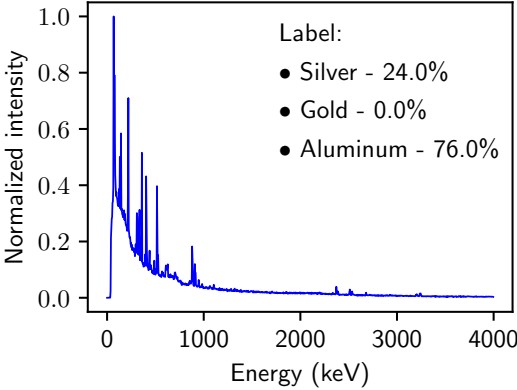

Figure 7: Muon spectroscopy data example. The dataset contains 10,000 samples, each with 1,000 data points. 6,000 samples are used for training, and 4,000 for testing.

### C.3 STOKES FLOW

Physics-informed machine learning (PIML) is a rapidly growing field (Karniadakis et al., 2021), primarily targeting forward and inverse problems involving PDEs. The key idea behind PIML is to incorporate the underlying physics or PDEs directly into the neural network, often through the loss function, to enhance accuracy and generalization while requiring less data. In this study, we investigate the effectiveness of KGI within the PIML framework.

Following Leng et al. (2024), we consider the 2-D Stokes flow in a square box with a moving lid, governed by the following system of PDEs and boundary conditions:

$$
\begin{aligned}
\mu \left( \frac{\partial^2 u}{\partial x^2} + \frac{\partial^2 u}{\partial y^2} \right) - \frac{\partial p}{\partial x} &= 0, & x \in (0,1), y \in (0,1); \\
\mu \left( \frac{\partial^2 v}{\partial x^2} + \frac{\partial^2 v}{\partial y^2} \right) - \frac{\partial p}{\partial y} &= 0, & x \in (0,1), y \in (0,1); \\
\frac{\partial u}{\partial x} + \frac{\partial v}{\partial y} &= 0, & x \in (0,1), y \in (0,1); \\
u(x,1) = u_1(x), v(x,1) &= 0, & x \in (0,1); \\
u(x,0) = v(x,0) = p(x,0) &= 0, & x \in (0,1); \\
u(0,y) = v(0,y) &= 0, & y \in (0,1); \\
u(1,y) = v(1,y) &= 0, & y \in (0,1),
\end{aligned}
\tag{14}
$$

where $\{u,v\}(x,y)$ denotes the velocity field, $p(x,y)$ is the pressure field, and $\mu$ is the dynamic viscosity, fixed at 0.01. The zero-pressure boundary condition at the bottom ($p(x,0) = 0$) is imposed solely to fix the constant component of $p$. Our goal is to learn an operator that maps the lid velocity $\hat{u}(x,0)$ to the full solution $\{u,v,p\}(x,y)$, as shown in Figure 8a.

The physics-informed DeepONet (Lu et al., 2021), designed for operator learning with universal approximation capabilities, is used as the architecture, as illustrated in Figure 8b. Both the branch and trunk networks consist of three hidden layers with 128 neurons each, using `Tanh` activation. For KGI, we employ the bias-modifying approach, setting the input domain $[a,b] = [-0.8, 0.8]$ for all the hidden layers, pre-initialized by Xavier normal. We sample 1200 instances of $\hat{u}(x,0)$ from a Gaussian process, using 1000 for training without the ground truth solutions. The L2 error relative to the ground truth for the remaining 200 instances serves as the metric for accuracy and generalization. Training is conducted over 50,000 iterations, each using a batch of 50 $\hat{u}(x,0)$ instances, with a learning rate of $10^{-3}$ for the Adam optimizer.

The results presented in Table 1 and Figure 4 demonstrate that with KGI, the PDE loss is reduced by a factor of three and continues to converge rapidly. The test error decreases from $\sim$18% to $\sim$10%. These improvements, achieved solely by initialization, are outstanding in the context of PIML, highlighting the strong potential of KGI in this field.

### C.4 DISENTANGLEMENT

A key characteristic of learned representations is disentanglement, where the latent space is structured by distinct, independent generative factors of variation. This structure not only reveals insights into the underlying data but also enables controlled manipulation of specific data attributes. One classic approach for achieving disentanglement is through VAE-based models (Higgins et al., 2017). These models regularize the latent space using the KL-divergence, encouraging the latent distribution to align with a standard normal distribution, thereby promoting latent independence. However, due to the non-convex nature of the loss function in VAE-based models, successful disentanglement is highly sensitive to initialization (Locatello et al., 2019). If an early training stage results in an entangled latent space, the model will struggle to separate generative factors later on.

In this experiment, we employ a VAE model with MLP backbones and `ReLU` activations on the XY dataset from Cha & Thiyagalingam (2023). The dataset consists of 33×33 images, each of size 64×64 and containing a circle with varying $x$ and $y$ positions. The encoder is composed of three linear layers with sizes 1024, 512, and 128, each using `ReLU` activations. This is followed by two separate layers of size two, which generate the mean and log-variance of the two latent

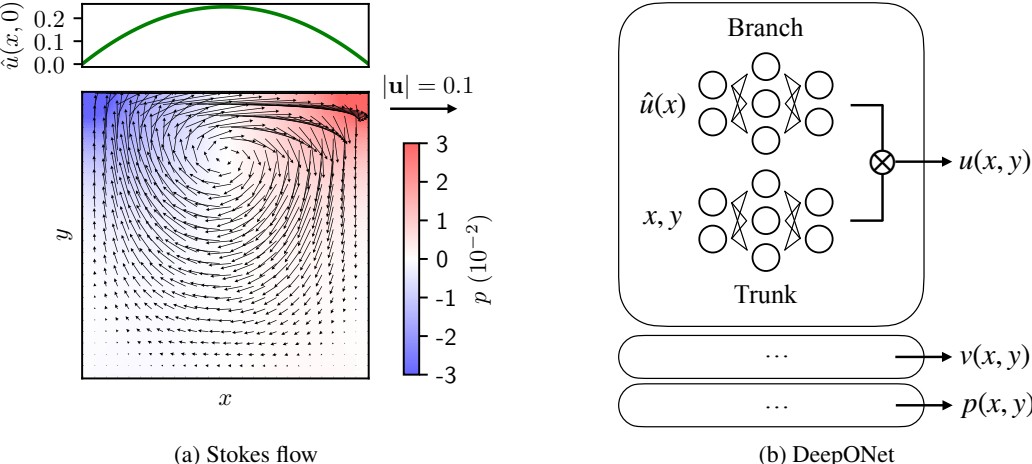

(a) Stokes flow

(b) DeepONet

Figure 8: Solving the 2-D Stokes flow in a square box with a moving lid using a physics-informed DeepONet. Panel (a) shows the input $\hat{u}(x, 0)$ in the top plot and the corresponding output fields $\{u, v, p\}(x, y)$ in the bottom plot. Panel (b) illustrates the use of three separate DeepONet modules, all with the same architecture, to predict the velocity and pressure fields. These predicted fields are then substituted into Eq. (14) to calculate the PDE loss for backpropagation, with the derivatives computed by automatic differentiation.

features. The decoder mirrors the encoder, followed by a `sigmoid` activation function to facilitate reconstruction. Training is performed for 1000 epochs with a batch size of 64, optimized by Adam with a learning rate of $5 \times 10^{-5}$. The VAE loss is the sum of the reconstruction loss and the KL-divergence from the latent distributions to the standard normal distribution. We aim to encode the images into independent latent variables, as shown in Figure 9.

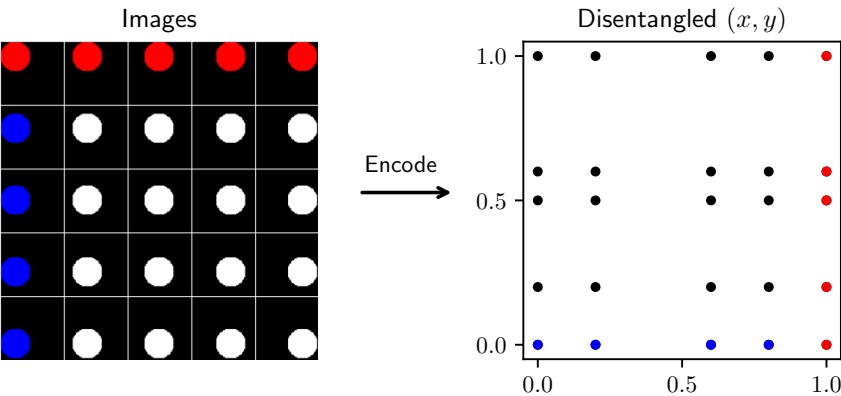

Figure 9: A perfectly disentangled latent space of the XY dataset (Cha & Thiyagalingam, 2023). The colored circles and their encoded positions demonstrate that perfect disentanglement does not necessitate ordering or monotonicity in latent variables. Additionally, the spacing between rows and columns in the latent space is not required to be uniform.

The models are pre-initialized using PyTorch's default initialization, which differs from He uniform by the absence of the $\sqrt{3}$ scaling factor. In applying KGI, we use the weight-modifying approach over the interval $[a, b] = [-0.1, 1.1]$ for the input layer, as the input images contain only values of zero and one. For the other hidden layers in the encoder, we set $[a, b] = [0.2, 0.8]$. For the layers responsible for generating the mean and log-variance, we calculate the minimum and maximum values of their inputs before training and use them as $a$ and $b$, since the final encoder layer has no activation function. To constrain the decoder, we similarly compute the minimum and maximum values of the reparameterized outputs before training and use them as $a$ and $b$. For the other hidden

layers of the decoder, we set $[a, b] = [0.49, 0.51]$. The reason disentanglement is facilitated by a narrow interval in the decoder remains unclear to us.

As shown in Table 1, KGI not only significantly improves the mean of JEMMIG but also reduces its variance. This indicates that KGI effectively addresses the non-convexity of disentanglement by enhancing local expressiveness. Figure 4 displays the JEMMIG history of the best models with and without KGI, both of which successfully achieve disentanglement, with KGI accelerating convergence. In the failed cases, JEMMIG remains mostly flat. Among the ten tested seeds, disentanglement was successfully achieved seven times with KGI and four times without it, based on a visual inspection of the JEMMIG history. The learned latent spaces corresponding to the best models with and without KGI are visualized in Figure 10.

No KGI
KGI

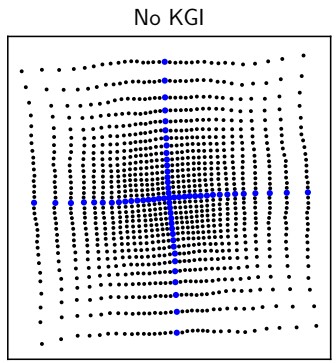 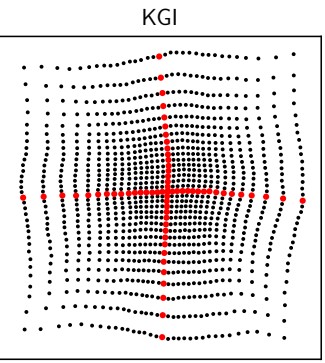

Figure 10: Disentangled latent features. The visualized models are those with the highest JEMMIG scores, one using KGI and the other without, corresponding to the two JEMMIG histories shown in Figure 4. Although these two cases appear similar, KGI increases the likelihood of achieving successful disentanglement as seen here.

## C.5 GPT-2 PRETRAINING

Large language models (LLMs) are gaining significant prominence, making them an ideal testbed for evaluating novel methodologies (Zhao et al., 2023). In this experiment, we pretrain GPT-2 from scratch (Radford et al., 2019). While a larger dataset like OpenWebText (Gokaslan & Gao, 2019), with eight million documents, is ideal for training convergence, resource constraints lead us to use the smaller Wiki-Text-2 dataset (Merity et al., 2017), which contains 40,000 documents. Therefore, the findings presented here primarily reflect KGI's impact during the early stages of pretraining. We train the model for 120 epochs, which correspond to approximately 45,000 steps of backpropagation, using a batch size of six and a learning rate of $5 \times 10^{-4}$.

GPT-2 includes five types of linear layers, which together account for the majority of its parameters:

1. `attn.attn` projects token embeddings into queries, keys, and values for self-attention.
2. `attn.proj` projects the output of self-attention back to the model's hidden size.
3. `mlp.fc` projects the output of self-attention into a higher-dimensional space.
4. `mlp.proj` projects the activated higher-dimensional output back to the hidden size.
5. `lm_head` maps the model's hidden states to vocabulary logits for language modeling.

To apply KGI, we assign the interval $[a, b] = [0, 1]$ for the `mlp.proj` layer, as its preceding activation function is GELU, while using $[a, b] = [-1, 1]$ for the remaining layers. Note that in Huggingface, GPT-2 implements all above linear layers (except `lm_head`) using `Conv1D`, which must be replaced with `nn.Linear` to apply KGI.

As shown in Figure 4, KGI leads to faster convergence of the training loss. While evaluation accuracy (presented in Table 1) shows a subtle improvement with KGI, we do not claim conclusive effectiveness because this experiment only covers the early stages of training, with perplexity remaining at the $10^4$ level. Still, it hints at the potential of KGI for improving the training dynamics of large-scale models, including LLMs.

