# OpenReview forum: "Enhancing Performance of Multilayer Perceptrons by Knot-Gathering Initialization"
_ICLR.cc/2025/Conference — ICLR 2025 Conference Withdrawn Submission_

### Official Review · Reviewer_n2kt · 2024-10-28

**Soundness:** 2
**Presentation:** 3
**Contribution:** 2
**Rating:** 3
**Confidence:** 4

**Summary:**

This manuscript focuses on the non-differentiability points called knots for understanding the expressivity of MLPs. The authors propose Knot Gathering Initialization (KGI), which ensures knots lie in the input domain. This objective is accomplished through initialization of the weight or bias of MLP.

**Strengths:**

Motivation and ideas are interesting. Indeed, several researchers have studied the expessivity of MLP in terms of the number of fragments in piecewise linearity. This study is in alignment with those studies, which deepen understanding of MLPs.

**Weaknesses:**

The major assumption and limitation of this manuscript is the [Weight-Act] pipeline, as described in Eq. 1. However, nowadays, it became standard to use normalization layers such as batch normalization or layer normalization with the [Weight-Norm-Act] pipeline. I think that the use of a normalization layer would yield different initialization methods in their theory, especially in Eqs. 1-6; in other words, I wonder that the proposed KGI with these equations would increase the number of knots as desired in practice.

For example, the normalization layers consist of a normalization step and an affine step that adopts gamma and beta parameters in the form of y=ReLU(z), z=gamma*zhat+beta, and zhat=normalize(Wx+b). For this scenario, I conjecture that initialization should be applied to gamma and beta, not the weight and bias. Indeed, the use of a normalization layer provides normal distribution, which can be exploited to sample xhat in place of Eq. 2.

In summary, there are plenty of practical and interesting theories that should be considered when adopting the normalization layer, which, however, is not discussed in the current manuscript. Not to mention that there are numerous architectures, such as skip connection or self-attention, which are not sufficiently targeted for the subject of KGI in this manuscript. I found that Section 5.1 provides some discussion, such as variants of weight configuration and future works with normalization layers; nevertheless, the current manuscript still sticks to the [Weight-ReLu] pipeline without normalization layer. I evaluate that the current form of KGI seems not compatible with modern architecture, which uses normalization layers. Indeed, the authors targeted MLP for CIFAR-10; MLP for muon spectroscopy; MLP for DeepONet; MLP for VAE; and MLP for GPT-2. All these examples rather seem to indicate the limitation and archictectural restriction of KGI onto MLP without normalization layers.

Minor comment: template of ICLR 2024, not ICLR 2025, was used. In the 2025 template, there should be numbering for all lines. Although this issue might be minor, the violation of format should not be allowed in principle.

**Questions:**

See the weaknesses.

---

### Official Review · Reviewer_Njo3 · 2024-10-30

**Soundness:** 4
**Presentation:** 4
**Contribution:** 4
**Rating:** 10
**Confidence:** 4

**Summary:**

This paper proposes an effective method to enhance MLP performance by increasing the number of knots within a meaningful range. It begins by reviewing prior research that establishes a correlation between the complexity of an MLP and the number of knots. Building on the premise that more knots lead to higher performance, it develops derivative formulas that efficiently increase the number of knots during the network initialization stage. The application of this initialization strategy across various AI-related tasks demonstrates the utility of their method.

The paper is underpinned by robust theoretical support, meticulous formula derivation, and is substantiated by ample experimental evidence. The contribution of this work is significant.

**Strengths:**

The paper is supported by solid theory, meticulous formula derivation, and extensive experimental validation.

In the related work section, it comprehensively details the necessary background and smoothly transitions to the objective of enhancing MLP performance by increasing knot density in meaningful input spaces.

Equations 3 and 4 detail the methods for increasing the number of knots within a specific range, while Equation 7 outlines how to determine this range. Equation 6 discusses how to balance this method with other initialization strategies.

The experimental section first validates that this method effectively increases the number of knots in 1- and 2-dimensional spaces, and then demonstrates improved performance across various AI-related tasks.

**Weaknesses:**

Section 3.2 states, “our objective is to increase, rather than to maximize, knot density within the input domain.” The rationale for not maximizing knot density could be better explained, potentially with supporting experimental data.

Equation 6 introduces a hyperparameter to balance this method with other initialization strategies. It would be beneficial to explore how this hyperparameter impacts MLP performance, ideally through comparative experiments.

**Questions:**

I concur with the notion that this method could be applied during the training stage as well, and I am eager to see its implementation in future research.

---

### Official Review · Reviewer_mQMq · 2024-11-03

**Soundness:** 2
**Presentation:** 2
**Contribution:** 2
**Rating:** 3
**Confidence:** 4

**Summary:**

The paper proposes a novel weight initialisation technique termed “Knot Gathering Initialization” (KGI) that attempts to increase the “local expressiveness” of MLPs through the concentration of “knots” in the relevant input domain. For ReLU-like activation functions, “knots” can be understood as the points of non-differentiability ($z=0$) giving rise to the piecewise linear approximation of (high-dim.) functions or decision boundaries. More broadly, the concept focuses on the “points” where (smooth) activation functions are most expressive. The paper's main contribution is the characterisation of KGI for MLPs, i.e. how to increase knot density in the most relevant region without adding significant computational overhead. Experimentally, improved convergence speed to lower losses compared to standard MLP initialisation is illustrated. Experiments include synthetic curve and surface fitting tasks, and more challenging tasks on image classification, physics modelling, disentanglement, and language modelling (considering GPT-2 and focusing on the linear layers). Extensions to other architectures and related initialisation and normalisation techniques are discussed.

**Strengths:**

The fundamental contribution of this paper is to identify a lightweight strategy for initialising MLP weights such that they lie in relevant regions, facilitating the fitting of the training objective more successfully. This is intimately connected to other approaches for weight initialisation, normalisation of activations, and, more broadly, dynamic activation functions, and the proposed KGI adds an original and elegant idea in that context. KGI can be significant for neural network training because it is computationally inexpensive and has the potential to improve convergence speed. In particular, I see two main strengths:

__S1.__ The paper effectively demonstrates that extending MLPs (or linear layers in models such as transformers) can (i) enhance the convergence to low-loss solutions and (ii) improve test accuracy under a fixed optimisation budget (like a set number of optimisation steps).

__S2.__ The authors address cases where the choice of initialisation is particularly problematic, like disentanglement tasks. They show experimentally that KGI can alleviate these challenges.

**Weaknesses:**

I view the above contributions as relevant but consider the paper as not yet ready for publication. Firstly, some aspects concerning the clarity of the paper, especially section 4 on “Experiments”, require some additional refinement to strengthen the contribution. I cover these aspects in __W1__, __W3__ and questions __Q1__ and __Q2__. Secondly, in my opinion, the quality of the paper can be improved with respect to the positioning of the method and the empirical evaluation, which I elaborate on below. In more detail, I see the following main weaknesses:

__W1.__ _Relationship between expressiveness / universal approximation / capacity measures and “local expressiveness”/ initialisation / normalisation_: The paper positions itself as a work on the expressive power of neural networks, and the authors reference the claim that “_[t]he number of knots in piecewise linear neural networks has been explored as an intuitive and interpretable measure of their expressive power._” (page 9) I view these claims as slightly overstated and a potentially detrimental shift in the contribution's focus. I see the contribution as follows. The main idea of KGI can be viewed as an initial weight perturbation that mostly influences the following optimisation steps during training. An MLP without KGI should be as expressive as one with KGI, but might not converge as quickly (see W3.2). As the authors mention in the related work section, I believe the domains closest to their proposal lie in normalisation techniques, work on adaptative activation functions, and other weight initialisation approaches. Therefore, “local expressiveness” should mostly be understood in this context and related to optimisation aspects. In this direction, considering “optimisation budgets”/training steps and KGI remains underexplored, in my opinion (see W3.2 on this below). However, KGI does not affect the expressiveness of MLPs or the capacity of the modelled hypotheses class in any way.

__W2.__ _Impact compared to related approaches is difficult to assess_: While innovative, KGI is not be sufficiently distinguished from previous strategies. The statement that “_[s]ince KGI introduces a mechanism distinct from previous studies and is compatible with most existing techniques […], we do not use a static baseline in our experiments._” (page 6) should be supported with clear empirical evidence and comparisons. KGI’s goal of enhancing “local expressiveness” is similar to other weight initialisation and general normalisation techniques. Running the same experiments with an additional comparison to related approaches, e.g. LSUV (Mishkin & Matas, 2016) or batch normalisation, for instance, and e.g. different initialisation strategies would allow for a more conclusive assessment of the relevance and benefits of KGI. This is currently missing.

__W3.__ _Empirical evidence and interpretation_: Several aspects make the interpretation of the experimental results challenging. \
__W3.1.__ _Train error / test error / overfitting_: In section 4.1, it appears that the focus is on training MSE, while in section 4.2 in some cases test results are considered. It would be helpful to provide more clarity on the considered metrics and compare both training and test curves. In that context, the potential risk of overfitting due to increased knot density at initialisation (mentioned in the discussion) should be evaluated and discussed, too. \
__W3.2.__ _Comparison of non-converged results_: A major issue I currently see is that most of the losses have not converged, in particular for the “no KGI MLPs” (see Figures 4, 5, and 6). It appears that implicitly, a fixed number of optimisation steps or an “optimisation budget” is assumed and “enhanced local expressiveness” is mostly attributed to attaining lower losses at the same number of optimisation steps. However, consideration of the “optimisation budget” / training steps is currently missing. It could beneficial to justify the choice of the optimisation budget. Additionally, as the “no KGI MLPs” have not yet converged, the actual improvement in general error or accuracy is not assessed, but rather the error or accuracy after a particular number of optimisation steps. One could expect “no KGI MPLs” to attain similar test errors or accuracies at a later stage of training. This is why I currently see the main improvements of KGI only concerning convergence speed. Extending the experiments to carefully cover converged results might provide additional insights and more rigorously evaluate the impact of KGI on final performance.

__W4.__ _Issues with the definition of “knots”_: The definition of “knots” is natural for ReLU activation functions but becomes increasingly fuzzy for other activation functions. Two issues related to section 3.3 on “Smooth activation functions” may illustrate this well. \
__W4.1.__ _Incorrect mutual information_: The displayed mutual information in Figure 2 (middle panel) is incorrect and likely stems from a frequent misconception when extending the notion of entropy to continuous random variables. In the derivation provided in Appendix A, it is stated that "[s]ince $Y = tanh(Z)$ is a deterministic transformation of $Z$, the conditional entropy $H_\mu (Y \vert Z)$ vanishes”. This statement is true when $Y$ and $Z$ are discrete random variables but false for continuous random variables. In the latter case, we need to consider the differential conditional entropy, which is $h(Y \vert Z)=-\infty$, leading to $I_\mu(Z;Y) = \infty$. \
__W4.2.__ _Mixed results for tanh_: tanh is highlighted as a particular example of a smooth activation function in section 3.3. However, the curve and surface fitting results illustrated in Figures 3e, 3f, 5d, and 6d show that KGI does not lead to consistent and substantially different results.

Minor remark:
Page 1, 4th paragraph. The authos might want to correct the sentence: “_We are generalizing the concept of knots to smooth activation functions (such as GELU and Tanh) is straightforward but rigorous._”


Based on the different aspects raised in W1 to W4, I view the submission as not yet ready for publication and suggest a major revision of the paper. However, I invite the authors to address my objections and clarify potential misunderstandings.

**Questions:**

Additionally, I have the following questions:

__Q1.__ _Figure 3, b and d_: In my opinion, it is not clearly stated what the different lines represent. The caption mentions that “[e]ach scenario includes ten model samples […]” (for b). Are these ten different initialisations (for the fixed width W and depth D)? An additional explanation of these plots might increase the clarity significantly here.

__Q2.__ _Figure 3, b_: If I understand correctly, the first and third plot in sub-figure b (i.e. on “No KGI, $W=5$” and “KGI, $W=5$”) compare the (average) number of knots within the input domains for an MLP with one hidden layer of width five and ReLU activations. As motivated in the paper up to this point (e.g. in section 3.3), activation $z=0$ is considered a knot. Wouldn’t this suggest that the maximum number of knots is five for the 1D curve problem if there are five ReLU activations, one at each hidden node/unit (and similarly 20 for the $W=20$ case)? As suggested in Q1, I believe a more careful explanation might be beneficial to increase clarity.

---

### Official Review · Reviewer_gNbj · 2024-11-04

**Soundness:** 3
**Presentation:** 4
**Contribution:** 2
**Rating:** 5
**Confidence:** 4

**Summary:**

Authors argue that, the density of "knots" (points of non-differentiability/big change) in the input domain is a proxy for an MLPs function approximation capacity. They design an initialization scheme, termed as knot gathering initialization (KGI) which initializes the weights in such a way that the number of knots in the input domain at the time of initialization is high. Through extensive experiments, authors show that KGI improves the training loss, test metrics and convergence speed in 6 different tasks - curve and surface fitting, image classification, time series regression, physics informed operation learning, representation disentanglement and large language model pertaining.

**Strengths:**

+ The paper is clearly written. I appreciate the effort authors have put into writing the paper.

+  The problem authors are tackling is intuitive. In the introduction section, it may help if authors could provide more "intuitive" reasoning on why knots are necessary. For example, discuss an extreme case where there are no knots in the input domain.

+ The initialization scheme is quite simple and straightforward, and explained well.

+ Experiments are comprehensive, covers multiple domains and architectures, and explained well.

+ Authors commitment to reproducibility and the release of code is also commendable.

**Weaknesses:**

+ **Is KGI helpful in truly deep networks?** I see that most networks used in the experiments are toyish. It has a maximum depth of 5, and the networks used in the real world data seems even shallower (disentangled uses 2 layered networks?). GPT is the only larger scale model explored, and here, the advantage seems very small. Can authors comment on if KGI still holds its advantage if the network is deep enough - let's say 15 layers or so. You can, for example, use  a Resnet style architecture for CIFAR10 (replacing conv layers with nn.Linear to support your framework). I do note that authors have mentioned that for sufficienctly sized models the final loss values are comparable, and only the convergence is faster - I'm trying to see if this holds for truly deep networks (for eg. the difference in slowness for GPT-2 is negligible and within confidence interval).

+ **What are the insights from experiments in Section B? How do the number of knots change as we train?** Can authorize summarize any patterns they are seeing across different network sizes and activation functions? Can authors also look at how the number of knots as we train? Are deeper networks able to increase their number of knots as training progresses? **Does more knots generally mean better training and test error?**

+ Authors outline in Section 5.1 that KGI is compatible with architecture as CNNs and GNNs. It will considerably strengthen the paper if authors can show that performing KGI on SoTA architectures in ImageNet etc improves performance.

Overall, while it is an interesting observation and useful contribution, I am not sure if KGI really moves the needle on practical application. If my concerns, particularly on KGI being valuable for sufficiently complex networks, is addressed, I am happy to increase my rating.

**Questions:**

I find the discussion in Section 3.3 a bit disorganized. Instead of a sound definition, it seems like z=0 is sort of assumed to be the knot point and then 3 different prospectives are chosen to justify this. Can authors, maybe, define knot point a bit formally for an arbitrary function f(x)?

In this statement
When random input Z ∼ N (μ, 1) is passed through Y = tanh(Z ), the mutual information (MI) (MacKay, 2003) between Z and Y is maximized at μ = 0, as detailed in Appendix A. This suggests that z = 0 is an optimal point for information transmission.

It is unclear to me what the significance of this statement is. Why was Z chosen to have this specific distribution?

---

### Official Review · Reviewer_qHPY · 2024-11-04

**Soundness:** 4
**Presentation:** 3
**Contribution:** 3
**Rating:** 5
**Confidence:** 3

**Summary:**

The proposes a new initialization method --- KGI --- based on the intuition that increasing the point of non-differentiability in the input domain increases the expressive power of neural networks. The proposed method uses weight perturbation such that knots are increased at the initialization. The method is evaluated on different datasets, although with smaller models, and shows improvement over the non-KGI baseline.

**Strengths:**

* The paper is well-written with clear motivation and the method is theoretically justified.
* The paper does an excellent job of covering the necessary background information required to understand the paper.
* Results across different domains show promising results.
* The proposed method not only shows improvement in performance but also faster convergence.

**Weaknesses:**

* Most of the experiments are shown using MLP; it is not clear if this will translate to CNNs/ViTs.

* With the current trend of foundational models, where the expressive power is increased by increasing the model parameters count, there is a possibility that KGI will not give performance gain. Further discussion on this would be beneficial.

*  In section 2, it is not clear how the input ranges are determined for different activation functions. Why is ReLU bounded between 0 and 1?

* In section 3.3, the use of mutual information does not make much sense. Since the transformation is deterministic, the MI will always be equal to the entropy H(Y). Why is the entropy maximized at the root of z=0. Further explanation would be helpful.

* There is no discussion of how increasing the knots improves the convergence rate and why this will not lead to overfitting.

* KGI can only be used when training a model from scratch/initialization and cannot be used fine-tuning/transfer learning.

* Accuracy for datasets is quite low as MLP is used. It is not clear if KGI will improvement when using larger and complex sota models.

**Questions:**

* >Training is performed for 30,000 epochs on unbatched data.

    Does it mean that the entire training data is fed into the model at once, without any mini-batches? Removing mini-batches will remove reduce the stochasticity/randomness for the baseline. KGI provides additional randomness (in eqn 6), and that might be a reason for improved performance and convergence instead of increased knots.

---

### Note · Authors · 2024-11-26

**Comment:**

Dear Program Chairs and Reviewers,

We sincerely thank the reviewers for their detailed and insightful comments. In particular, we deeply appreciate that all the reviewers recognized Knot Gathering, aimed at enhancing local expressiveness, as offering **a new perspective** to the community.

During the rebuttal stage, we conducted deeper theoretical evaluations and extended experiments based on the reviewers' comments. Based on our new understanding, we have decided to withdraw the paper for a major revision.

Nevertheless, we would like to share our findings below.

---

## Major Concerns
We believe there are two major concerns raised by the reviewers:

1. The current paper only focuses on MLPs.
2. Lack of insights into the relationship and interplay between Knot Gathering and feature normalization (such as batch normalization and layer normalization).

We consider the other concerns as minor and addressable through extended experiments and open discussions.

### First Concern
Regarding the first concern, we clarified in the paper that our focus on MLPs was to enable a thorough experimental evaluation. However, we also outlined how our method could be extended to other architectures, such as CNNs, GCNs, and LoRA. Moreover, MLPs are significant in their own right—for example, in transformers, all trainable layers, apart from embeddings and the two scalars in layer normalization, are linear layers. These linear layers account for the majority of the model's parameters. Notably, self-attention itself is not trainable; instead, the trainable components are the linear layers used in the interaction between multi-heads.

### Second Concern
The second concern is the primary reason for our withdrawal.

The reviewers noted a potential connection between Knot Gathering and feature normalization methods (batch, group, and layer normalization). Upon investigation during the rebuttal, we discovered that **normalization effectively gathers knots during training using our bias-modifying formula**. Specifically, our knot-sampling-based implementation of the bias-modifying formula is equivalent to Pre-Layer Norm (Pre-LN, as in [https://arxiv.org/pdf/2002.04745](https://arxiv.org/pdf/2002.04745), see Fig. 1), which has been proven effective in transformers. However, our weight-modifying formula has not been previously explored.

### Implications of this Discovery
* **Positive:** Knot Gathering offers an interpretable and visual mechanism for normalization—gathering knots of activation functions into the input domain to enhance local expressiveness. This can provide valuable insights into the mechanism of normalization, which, despite its empirical success, remains not fully understood. For example, "Although batch normalization has become popular due to its strong empirical performance, the working mechanism of the method is not yet well understood" (Wikipedia).
* **Negative:** To reflect this new finding, the paper requires significant revisions that follow a different chain of thought, which cannot be accommodated within the current timeframe. Continuing with minor revisions and responses to reviewers while disregarding these insights would not align with ICLR's Code of Ethics.

Therefore, we plan to rewrite the paper with the current and additional experiments under a revised title: *Knot Gathering: How Feature Normalization Works and Improves*, which more accurately reflects the key novelty of this work.

---

Once again, we sincerely thank the reviewers and program chairs for their time and thoughtful feedback.

Best regards,
Authors

**Withdrawal Confirmation:**

I have read and agree with the venue's withdrawal policy on behalf of myself and my co-authors.